# Effect of gas exchange data selection methods on resting metabolic rate estimation in young athletes

Victor Zaban Bittencourt[1], Raul Freire[2], Juan M. A. Alcantara [3,4,5], Luiz Lannes Loureiro[1], Taillan Martins de Oliveira[1], Fábio Luiz Candido Cahuê [1], Alex Itaborahy[1], Anna Paola Trindade Rocha Pierucci [1] *

1 DAFEE Laboratory, Graduate Program in Nutrition, Instituto de Nutrição Josué de Castro, Universidade Federal do Rio de Janeiro, Rio de Janeiro, Brazil, 2 Olympic Laboratory, Brazil Olympic Committee, Rio de Janeiro, Brazil, 3 Institute for Innovation & Sustainable Food Chain Development, Department of Health Sciences, Public University of Navarre, Campus Arrosadía, Pamplona, Spain, 4 Navarra Institute for Health Research, IdiSNA, Pamplona, Spain, 5 Centro de Investigación Biomédica en Red Fisiopatología de la Obesidad y Nutrición (CIBERobn), Instituto de Salud Carlos III, Madrid, Spain

* pierucci@nutricao.ufrj.br

**Data Availability Statement:** All relevant data are within the paper and its Supporting information files.

## Abstract

This cross-sectional study analysed the effect of the gas exchange data selection methods on the resting metabolic rate (RMR) estimation and proposed a protocol shortening providing a suitable RMR estimation for young athletes. Sixty-six healthy young Brazilian athletes performed a 30-minute RMR assessment. Different methods of gas exchange data selection were applied: short and long-time intervals, steady-state (SSt), and filtering. A mixed one-way ANOVA was used to analyse the mean differences in gas exchange, RMR, respiratory exchange ratio (RER), and coefficients of variation across all methods. Additionally, paired Student's t-test were used to compare the first and best SSt RMR values for each SSt method (3, 4, and 5-min). The 5-min SSt method provided the lowest RMR estimate (1454 kcal.day$^{-1}$). There was a statistical difference between methods (F = 2.607, p = 0.04), but they presented a clinically irrelevant absolute difference (~36 kcal.day$^{-1}$). There were no differences in RER among methods. In addition, using the SSt method, 12 minutes of assessment were enough to obtain a valid estimation of RMR. The 5-min SSt method should be employed for assessing the RMR among young athletes, considering the possibility of obtaining a shortened assessment (~12 min) with an acceptable and low coefficient of variation.

## Introduction

The resting metabolic rate (RMR) is the minimum energy required to maintain human physiological functions and homeostasis after fasting (commonly 12 h), at rest, in environmental thermoneutrality (ambient temperature 22-25˚C), and when the subject is awake [1, 2]. Proper RMR assessment is essential during the lifecycle, especially in adolescence. This period is marked by rapid pubertal growth and development that promotes physiological changes increasing RMR

**Funding:** This research received funding FAPERJ - Fundação Carlos Chagas Filho de Amparo à Pesquisa do Estado do Rio de Janeiro, E-26/201.042/2021 Dr Anna Paola Trindade Rocha Pierucci and MCIN/AEI FJC2020-044453-I Juan M. A. Alcantara.

**Competing interests:** NO authors have competing interests.

[3]. Thus, RMR assessment in young is relevant for developing an individualised nutritional plan to ensure suitable energy needs [4, 5]. Athletes, even adolescents, have higher energy needs than sedentary individuals because exercise training promotes greater energy demands [6, 7]. Therefore, if the energy needs determination is incorrect, it can induce hormonal dysregulation, total body weight and fat-free mass reduction, and loss of muscular strength and/or physical performance [8]. Notably, these dysregulations may be aggravated in adolescent athletes, whose energy requirements may fluctuate more than in non-adolescent athletes [9].

Indirect calorimetry (IC) is considered the gold standard technique to accurately assess the RMR. IC estimates whole-body energy requirements based on the measured volumes of oxygen consumption ($VO_2$) and carbon dioxide production ($VCO_2$) [10–12] The RMR assessment generally lasts from 10 to 30 min [13], and it is widely recommended to discard the initial 5-min period data [1, 14]. Moreover, from the entire assessment (e.g., 30 min), a shorter period must be selected and "used" for estimating the RMR. However, there is no consensus on the "best method" for selecting the gas exchange data and then estimating the RMR [15]. Thus, different gas exchange data selection methods, such as Steady State (SSt), time interval (TI), and filtering, have been proposed in the literature [15–18].

Generally, SSt (concretely the 5-min SSt method), which is defined as the period that presents a coefficient of variation (CV) <10% for $VO_2$ and $VCO_2$ and <5% for respiratory exchange ratio (RER) is supposed to be the method that provides the "best estimate" of RMR [19]. However, the study supporting this hypothesis was performed in ventilated intensive care unit patients [19], and for healthy subjects, the recommendations are still inconsistent [18]. Beyond SSt, other methods, such as TI and filtering, may be used [16, 18]. Briefly, the TI method selects the gas exchange data from the entire RMR assessment for pre-defined time intervals (ranging from 5 to 20-min). Lastly, the filtering method involves selecting the gas exchange data based on specific pre-defined "cut-off points", which vary with the "intensity" of the filter applied. In this regard, three different filters have been proposed: (i) the low filter, which selects values between 85 and 115% of the average RMR; (ii) the medium filter, which selects values between 90 and 110% of the average RMR; and (iii) the high filter, which selects the values between 95 and 105% [18].

Although the comparison between the methods in healthy individuals has been debated previously, to the best of the authors' knowledge, there is still no recommendation on the best method to select the gas exchange and estimate the RMR in young athletes. Moreover, given that it is known that VO2 and VCO2 kinetics could be influenced, among other factors, by age [20], determining the method for gas exchange data selection that could provide most accurate RMR estimations deserves attention. In addition, it remains unclear whether a shortened protocol can be employed to reduce the duration of the RMR assessment, from the traditional 30-min assessment, for this specific population. Previous literature in a sample of older high-level athletes [16] suggests that a 20-min measurement (without an acclimation period prior to the RMR assessment) provides a valid RMR estimation. Regrettably, whether this reduced protocol can be used in young athletes remains elusive.

Thus, this study aimed to compare the effect of different gas exchange data selection methods to estimate the RMR and to propose a reduced RMR protocol in a cohort of young athletes.

## Methods

### Participants

This cross-sectional study included 66 athletes (30 females) from five different sports [judo (n = 21), synchronised swimming (n = 20), swimming (n = 13), and water polo (n = 12)]. The

ages ranged from 12 to 18 years. The recruiting period was between September and November 2017. The inclusion criteria were: a) training volume of at least 20 h/week, b) practice for at least one year before recruitment, and c) competed in at least one official national championship during the last season. The experimental procedures and possible risks associated with the study were explained to the participants and their parents (or legal guardians), and written informed consent was obtained before enrolment. The Ethical Committee on Research approved this study for Humans at the Hospital University of Rio de Janeiro Federal University (CAEE: 58179716.3.0000.5257).

## Body composition and anthropometrical assessment

Body mass (kg) and height (cm) were measured using a scale with a stadiometer (Ramuza, São Paulo, Brazil). Body fat percentage (%), fat-free mass (FFM, kg), and fat mass were determined using GE Healthcare's Lunar dual-energy X-ray absorptiometry (DXA; Encore 2008, software version 12.3).

## Resting metabolic rate assessment

Before the test, the participants were instructed to avoid consuming any thermogenic supplements [21–23], sleep or appetite inhibitors [24], or other substances known to affect RMR during the preceding 24 h. In addition, they were instructed to fast for 8–12 hours and avoid any exercise training during the preceding 48 h [14]. All tests were performed in the morning (07:00–10:00). On the test day, athletes were picked up at home by a driver provided by the research staff to avoid physical activity before testing. Individual interviews were conducted on arrival to verify whether they had completed the previous instructions. Before the RMR assessment, the athletes rested quietly in a recliner (supine position) for 30 minutes and were instructed about the procedures concerning the assessment (e.g. not sleep, breathe normally).

The Vmax Encore 29 System metabolic cart (VIASYS Healthcare Inc., Yorba Linda, CA), equipped with a plastic canopy, was used for the IC assessment. The equipment was calibrated for gas analysers (using standard concentration gas bottles) and flow (using a 3 L syringe) before each measurement, following the manufacturer's instructions.

**Gas exchange data selection methods.** Oxygen uptake ($VO_2$) and carbon dioxide production ($VCO_2$) were continuously collected every 1 min (one sample per minute) for 30 minutes. According to current guidelines, the first 5 minutes of measurement data were discarded [1, 14]. The remaining 25-min period was further processed using different gas exchange data selection methods, as previously described [1, 2]: (i) short and long TI; (ii) SSt; and (iii) filtering methods.

Briefly, for short TI methods, $VO_2$ and $VCO_2$ values were averaged into five fixed 5-min windows, including 6–10 min, 11–15 min, 16–20 min, 21–25 min, and 26–30 min. Subsequently, $VO_2$ and $VCO_2$ values were averaged using the data within the 6–25 min and–6–30 min windows for the long TI methods. For the SSt method, $VO_2$ and $VCO_2$ values were averaged into different window lengths (3, 4, 5, and 10-min). Then, the period that accomplished the SSt criteria (i.e., presented a CV for $VO_2$ and $VCO_2$ <10% and for RER <5%) and the lowest CV average (i.e., presenting the lowest CV for $VO_2$, $VCO_2$, and RER average; from now on named as mean CVs) was selected and used for further analyses. Finally, for the filtering method, the average of the RMR during the last 25-min was calculated (AverageRMR). Afterward, three intensities of filters (Low–85–115%, Medium–90–110%, and High–95–105% of AverageRMR) were applied. Only those values that fell within the range of each filtering "intensities" were used for further analyses. Additionally, the first and best SSt were computed as previously proposed by Alcantara et al (2020) [18]. The "First SSt" consisted of the first SSt

period of 3, 4, or 5-min that accomplished the CV criteria (as mentioned above), and the "Best SSt" was represented as the SSt period of 3, 4, or 5-min that acinous the CV criteria and presented the lowest mean CV. In addition, the cumulative number of subjects (expressed as %) who achieved the first and best SSt periods was calculated. Finally, RMR values were estimated using Weir's equation, considering no urinary nitrogen excretion [25].

### Statistical analysis

A one-way repeated-measures analysis of variance (ANOVA) was employed to analyse the mean differences in RMR, $VO_2$, $VCO_2$, RER, and coefficients of variation ($CV\_VO_2$, $CV\_VCO_2$, $CV\_RER$, and mean CVs) across the gas exchange data selection methods (i.e., TI, SSt [Best SSt], and filtering), followed by Tukey's post-hoc test when appropriate. For ANOVA analyses, partial eta squared ($\eta^2$) were calculated and interpreted as 'small' (S, $\eta^2$ ranged from 0.2 to 0.5), 'medium' (M, $\eta^2$ ranged from 0.51 to 0.80), 'large' (L, $\eta^2 > 0.80$) effects. The ANOVA was chosen because not all methods returned valid values for all subjects (i.e., accomplished the SSt and/or the filtering methods criteria; we observed that TI 6-10min and 5min SSt reported 65 valid values [98,5%] and 10min SSt reported 51 [77,3%]). Additionally, a paired Student's t-test was used to compare the first and best SSt RMR values for each SSt method (3, 4, and 5-min length). Cohen's d test was used to calculate the effect size for paired t-test and interpreted as 'Small' (S, $d = 0.2$–0.5), 'Medium' (M, $d = 0.5$–0.8), and 'Large' (L, $d > 0.8$) [26].

Simple and multiple linear regressions were used to analyse the variance in RMR explained using different gas exchange data selection methods according to classical RMR determinants (body mass, fat-free mass, fat mass, and sex) [27]. Simple linear regression was used when body mass was selected as the independent variable (Model 1). For multiple linear regression models, sex and body mass (Model 2) and sex and body composition (fat-free mass and fat mass; Model 3) were selected as independent variables.

Statistical analyses were performed using GraphPad Prism version 9.2.0 (GraphPad Software, San Diego, California, USA), with a significance level fixed at p<0.05. All data are reported as mean ± standard deviation (SD), unless otherwise stated.

## Results

Table 1 depicts the anthropometric and body composition characteristics of the participants in the present study.

### Differences among gas exchange data selection methods

Fig 1 shows the RMR, $VO_2$, $VCO_2$, and RER for each method. There was a significant difference across methods for RMR estimations (F = 2.607, $p = 0.04$; Fig 1A) and $VO_2$ (F = 2.548,

**Table 1. Age, anthropometric, and body composition variables of male and female adolescent athletes.**

|  | Total (n = 66) [95% CI] | Males (n = 36) [95% CI] | Females (n = 30) [95% CI] |
|---|---|---|---|
| Age (years) | 14.8 ± 2.2 [13–15] | 14.8 ± 2.2 [13–15] | 14.8 ± 2.3 [13–17] |
| Body Mass (kg) | 60.06 ± 12.9 [56.1–62] | 60.0 ± 15.2 [57.7–67.2] | 60.0 ± 8.7 [53.2–59.4] |
| Height (cm) | 163.8 ± 9.4 [161–166] | 164.0 ± 11.2 [165–172] | 163.8 ± 4.6 [157–163] |
| Body Fat (%) | 25.2 ± 8.0 [20.5–27.6] | 20.2 ± 7.0 [14.6–21.0] | 30.2 ± 5.2 [26.5–31.5] |
| Fat-Free Mass (kg) | 43.94 ± 10.5 [38.3–46.2] | 44.2 ± 11.7 [45.9–53.3] | 43.8 ± 5.4 [36.2–40.7] |

Data are shown as mean ± standard deviation (SD), and as 95% confidence interval (CI).

$p = 0.04$; Fig 1C). Post-hoc analyses showed that the RMR retrieved by the 5-min SSt method was lower than that retrieved by the 6–10 min ($p = 0.001$) and both long TI methods (i.e., 6–25 min [$p = 0.02$] and 6–30 min, [$p = 0.03$]). Furthermore, the RMR retrieved by the 6–30 min TI method was higher than that retrieved by the low filter method ($p = 0.03$). Although we found statistical differences between the gas exchange data selection methods ($p = 0.04$), it could be considered clinically irrelevant since the greater mean difference was 46 kcal.day$^{-1}$. There were no significant mean differences in the VCO$_2$ ($p = 0.07$; Fig 1D) and RER ($p = 0.19$; Fig 1B) across methods. Finally, among the different gas exchange data selection methods, the 5min SSt method yielded the lowest RMR estimation (1454 ± 282 kcal.day$^{-1}$), while all methods provided similar RER estimates (0.81–0.85).

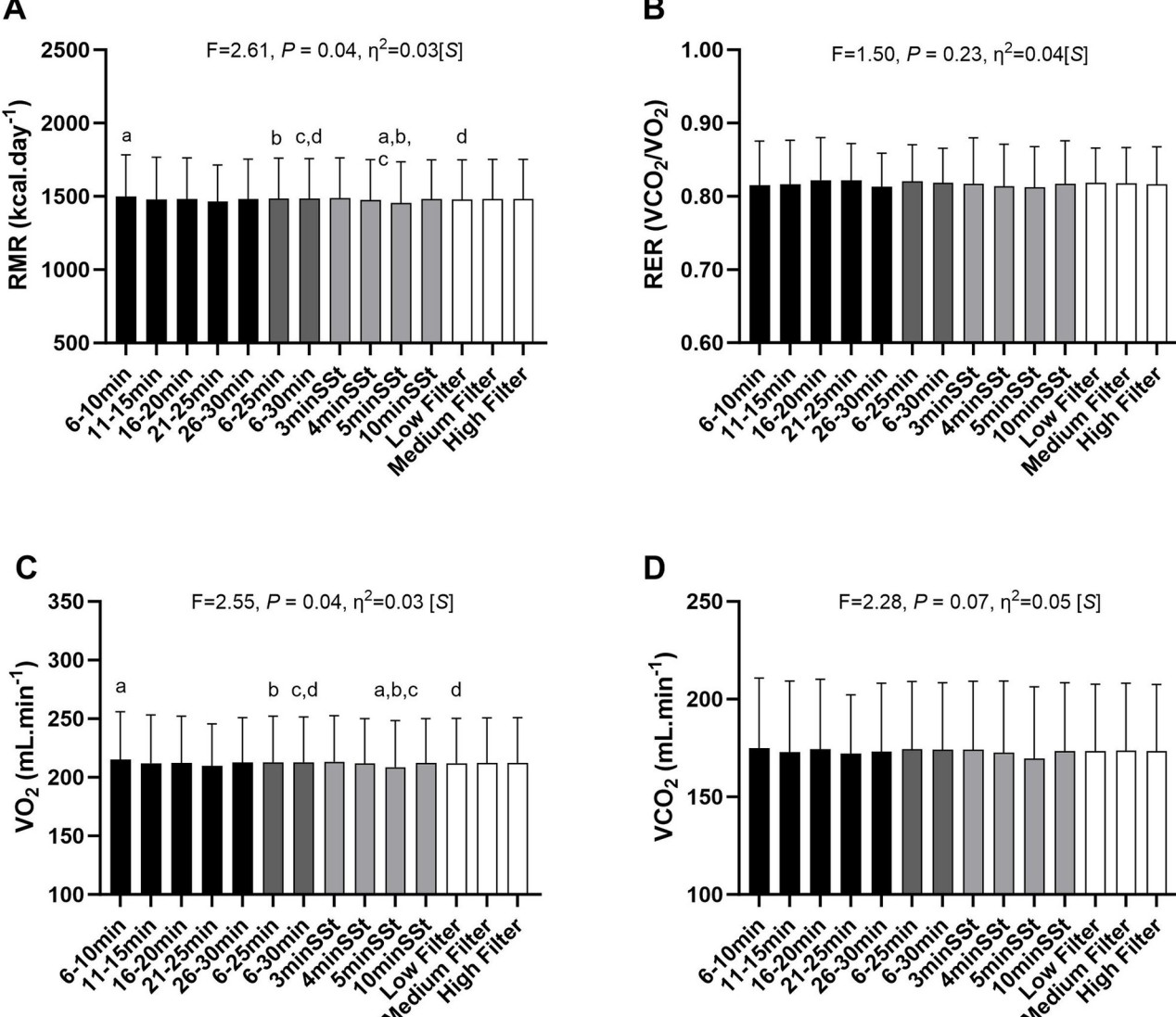

**Fig 1. Differences among gas exchange data selection methods for RMR (A), VO2 (B), VCO2 (C), and RER (D) across the different methods.**
Black columns represent the short Time Interval (TI) methods; Dark grey columns represent the long TI methods; Light grey columns represent the Steady-State (SSt) methods; White columns represent the Filtering methods. P-values come from the One-Way repeated measures analysis of variance. Data are presented as mean ± standard deviation (SD). Effect size: S—Small, M—Medium, L—Large. Min—minutes; VO2—volume of oxygen consumption; VCO2—volume of carbon dioxide production; RMR—resting metabolic rate; RER—respiratory exchange ratio.

## Coefficient of variation yielded by each method

Fig 2 presents the coefficient of variation (CV, expressed as a percentage) for $VO_2$ (Panel A), $VCO_2$ (Panel B), RER (Panel C), and mean CVs (Panel D) for each gas exchange data selection method. As expected by design (i.e., SSt methods use the CV for selecting the gas exchange data), all the SSt methods showed that all individuals had acceptable CV values on all parameters. In most methods, 75% of the subjects (3$^{rd}$ quartile) were below the recommended CV limits (i.e., <10% for $VO_2$ and $VCO_2$ and <5% for RER, see mean ± SD), indicating that they met

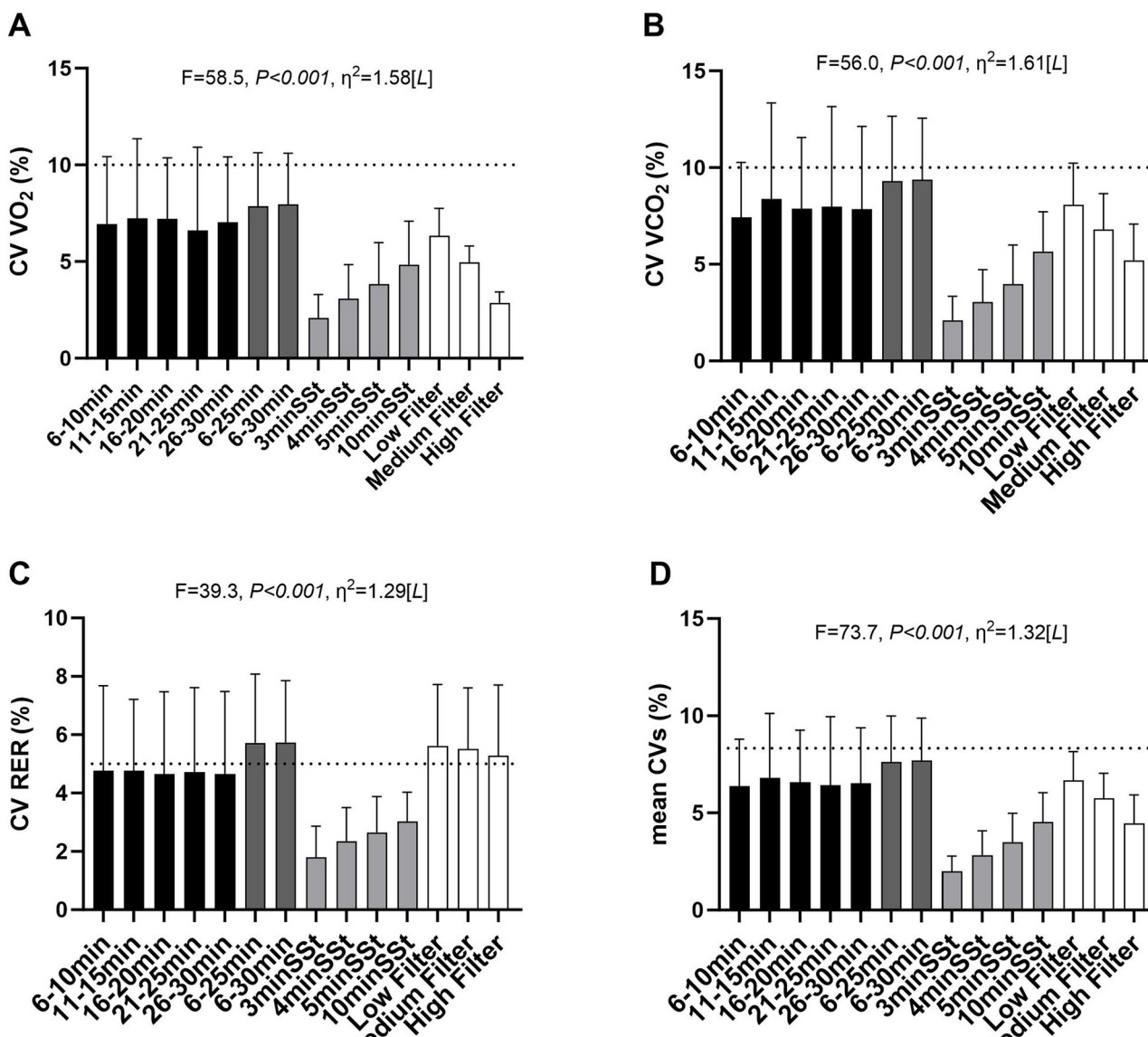

**Fig 2. Coefficients of variation (CV) of VO$_2$ (A), VCO$_2$ (B), RER (C), and mean of the coefficients of variation (Mean CVs) (D).** The dotted line represents the accepted variation limit for each variable, 10% for VO$_2$ and VCO$_2$, and 5% for RER. Black columns represent the short Time Interval (TI) methods; Dark grey columns represent the long TI methods; Light grey columns represent the Steady-State (SSt) methods; White columns represent the Filtering methods. P-values come from the One-Way repeated measures analysis of variance. Data are presented as mean ± standard deviation (SD). Effect size: S—Small, M—Medium, L—Large. Min—minutes; VO$_2$—volume of oxygen consumption; VCO$_2$—volume of carbon dioxide production; RER—respiratory exchange ratio.

the criteria for a "valid" RMR assessment (i.e., the assessment accomplished the gas exchange stability criteria).

## Variance in RMR explained by independent variables

Model 1 (i.e., body mass was included as the independent variable) explained ~47% ($R^2$ ranged from 0.37–0.52 among all methods, see Table 2) of the variance in the RMR and presented a SEE of approximately 202 kcal.day$^{-1}$. When sex was added as a predictor (Model 2), the explained variance increased slightly compared to Model 1, approximately 63% ($R^2$ 0.59–0.66 among all methods and retrieved a SEE of ~168 kcal.day$^{-1}$ (Table 3). Finally, in the model that included sex and body composition variables (FFM and FM) as predictors (Model 3), there was no significant improvement in RMR prediction compared to the previous model (Table 4).

## Comparison between the first and best steady-state periods of assessment

Fig 3 shows the cumulative percentage of subjects who achieved the first and best SSt using different period lengths (3, 4, and 5-min SSt). We observed that more than 80% of the subjects presented the first SSt between the 8th and 10th, and 8th and 11th min of assessment for the 3 and 4 min SSt methods, respectively (Fig 3A and 3B), while the Best SSt for the same percentage of subjects (80%) was reached later (~24–26 min). Moreover, for the 3 and 4 min SSt methods, there were no significant differences between the mean RMR obtained considering the first *vs* the best SSt ($p = 0.172$, and p = 0.105; Fig 3A and 3B, respectively). Concerning the 5-min SSt method, a slightly longer time (~15 min; Fig 3C) was needed to achieve the SSt criteria by 80% of the subjects. Moreover, statistically ($p < 0.001$) but not clinically relevant differences (~36 kcal.day$^{-1}$) were observed between the mean RMR values considering the first *vs* the best SSt (Fig 3C).

**Table 2. Variance in resting metabolic rate (RMR) explained by weight, as the independent variable, in each of the different methods for gas exchange data selection.**

| Methods of data selection | Model $R^2$ | Constant | Body Mass | | SEE |
|---|---|---|---|---|---|
| | | | β | *p* | (kcal.day$^{-1}$) |
| TI 6–10 min | 0.52 | 553 | 15.8 | <**0.0001** | 201 |
| TI 11–15 min | 0.49 | 528 | 15.8 | <**0.0001** | 207 |
| TI 16–20 min | 0.44 | 608 | 14.6 | <**0.0001** | 212 |
| TI 21–25 min | 0.37 | 724 | 12.4 | <**0.0001** | 199 |
| TI 26–30 min | 0.42 | 663 | 13.7 | <**0.0001** | 209 |
| TI 6–25 min | 0.49 | 577 | 15.1 | <**0.0001** | 198 |
| TI 6–30 min | 0.49 | 595 | 14.8 | <**0.0001** | 197 |
| 3-min SSt | 0.43 | 640 | 14.1 | <**0.0001** | 209 |
| 4-min SSt | 0.47 | 595 | 14.7 | <**0.0001** | 201 |
| 5-min SSt | 0.52 | 503 | 15.8 | <**0.0001** | 198 |
| 10-min SSt | 0.44 | 624 | 14.4 | <**0.0001** | 203 |
| Low Filter | 0.48 | 600 | 14.6 | <**0.0001** | 197 |
| Medium Filter | 0.48 | 603 | 14.6 | <**0.0001** | 198 |
| High Filter | 0.48 | 598 | 14.7 | <**0.0001** | 196 |

Unstandardized β (beta) and *p* values (significant values in bold) from simple linear regression; SEE—Standard Error of the Estimate.

**Table 3. Variance in resting metabolic rate (RMR) explained by sex and weight, as the independent variables, in each of the different methods for gas exchange data selection.**

| Methods of data selection | Model $R^2$ | Constant | Sex | | Body Mass | | SEE |
|---|---|---|---|---|---|---|---|
| | | | β | p | β | p | (kcal.day$^{-1}$) |
| TI 6–10 min | 0.65 | 542 | 216.6 | **<0.0001** | 14.0 | **<0.0001** | 171 |
| TI 11–15 min | 0.63 | 520 | 221.2 | **<0.0001** | 14.0 | **<0.0001** | 177 |
| TI 16–20 min | 0.59 | 600 | 222.6 | **<0.0001** | 12.7 | **<0.0001** | 182 |
| TI 21–25 min | 0.60 | 707 | 239.0 | **<0.0001** | 10.6 | **<0.0001** | 160 |
| TI 26–30 min | 0.61 | 654 | 242.4 | **<0.0001** | 11.6 | **<0.0001** | 172 |
| TI 6–25 min | 0.65 | 569 | 225.3 | **<0.0001** | 13.2 | **<0.0001** | 165 |
| TI 6–30 min | 0.65 | 587 | 228.2 | **<0.0001** | 12.9 | **<0.0001** | 163 |
| 3-min SSt | 0.63 | 631 | 251.8 | **<0.0001** | 12.0 | **<0.0001** | 169 |
| 4-min SSt | 0.61 | 587 | 209.4 | **<0.0001** | 12.9 | **<0.0001** | 173 |
| 5-min SSt | 0.66 | 487 | 217.6 | **<0.0001** | 14.1 | **<0.0001** | 167 |
| 10-min SSt | 0.65 | 548 | 254.4 | **<0.0001** | 13.0 | **<0.0001** | 162 |
| Low Filter | 0.65 | 592 | 225.5 | **<0.0001** | 12.7 | **<0.0001** | 163 |
| Medium Filter | 0.64 | 595 | 227.1 | **<0.0001** | 12.7 | **<0.0001** | 165 |
| High Filter | 0.65 | 590 | 228.7 | **<0.0001** | 12.8 | **<0.0001** | 162 |

Unstandardized β (beta) and *p* values (significant values in bold) from multiple linear regression; Sex—1 = men; 0 = women; SEE—Standard Error of the Estimate.

## Discussion

This study aimed to compare different gas exchange data selection methods proposed in the literature (TI, SSt, and filtering) to estimate RMR and verify the reliability of a shortened RMR protocol for healthy young athletes. A shorter analysis protocol would reduce individual time spent inside the equipment, bringing more comfort during the exam besides improving the

**Table 4. Variance in resting metabolic rate (RMR) explained by sex, fat-free mass (LBM) and fat mass (FM), as the independent variables, in each of the different methods for gas exchange data selection.**

| Methods of data selection | Model $R^2$ | Constant | Sex | | FFM | | FM | | SEE |
|---|---|---|---|---|---|---|---|---|---|
| | | | β | p | β | p | β | p | (kcal.day$^{-1}$) |
| TI 6–10 min | 0.67 | 579 | 269.9 | **<0.0001** | 11.4 | **<0.0001** | 19.0 | **<0.0001** | 168 |
| TI 11–15 min | 0.64 | 546 | 245.3 | **<0.0001** | 12.9 | **<0.0001** | 16.0 | **<0.0001** | 176 |
| TI 16–20 min | 0.62 | 621 | 266.0 | **<0.0001** | 10.7 | **<0.0001** | 17.0 | **<0.0001** | 178 |
| TI 21–25 min | 0.61 | 723 | 270.2 | **<0.0001** | 9.3 | **0.0001** | 13.4 | **0.0004** | 160 |
| TI 26–30 min | 0.62 | 678 | 276.4 | **<0.0001** | 10.0 | **<0.0001** | 14.7 | **0.0002** | 171 |
| TI 6–25 min | 0.67 | 596 | 266.3 | **<0.0001** | 11.3 | **<0.0001** | 17.1 | **<0.0001** | 162 |
| TI 6–30 min | 0.67 | 613 | 267.8 | **<0.0001** | 11.1 | **<0.0001** | 16.6 | **<0.0001** | 160 |
| 3-min SSt | 0.65 | 656 | 290.3 | **<0.0001** | 10.2 | **<0.0001** | 15.6 | **<0.0001** | 167 |
| 4-min SSt | 0.62 | 614 | 242.9 | **<0.0001** | 11.4 | **<0.0001** | 15.9 | **<0.0001** | 172 |
| 5-min SSt | 0.67 | 519 | 241.1 | **<0.0001** | 12.9 | **<0.0001** | 16.0 | **<0.0001** | 167 |
| 10-min SSt | 0.66 | 567 | 282.9 | **<0.0001** | 11.6 | **<0.0001** | 16.2 | **0.0001** | 160 |
| Low Filter | 0.67 | 618 | 267.3 | **<0.0001** | 10.8 | **<0.0001** | 16.7 | **<0.0001** | 160 |
| Medium Filter | 0.66 | 621 | 270.3 | **<0.0001** | 10.7 | **<0.0001** | 16.8 | **<0.0001** | 161 |
| High Filter | 0.67 | 617 | 268.4 | **<0.0001** | 10.9 | **<0.0001** | 16.5 | **<0.0001** | 159 |

Unstandardized β (beta) and *p* values (significant values in bold) from multiple linear regression; Sex—1 = men; 0 = women; SEE—Standard Error of the Estimate.

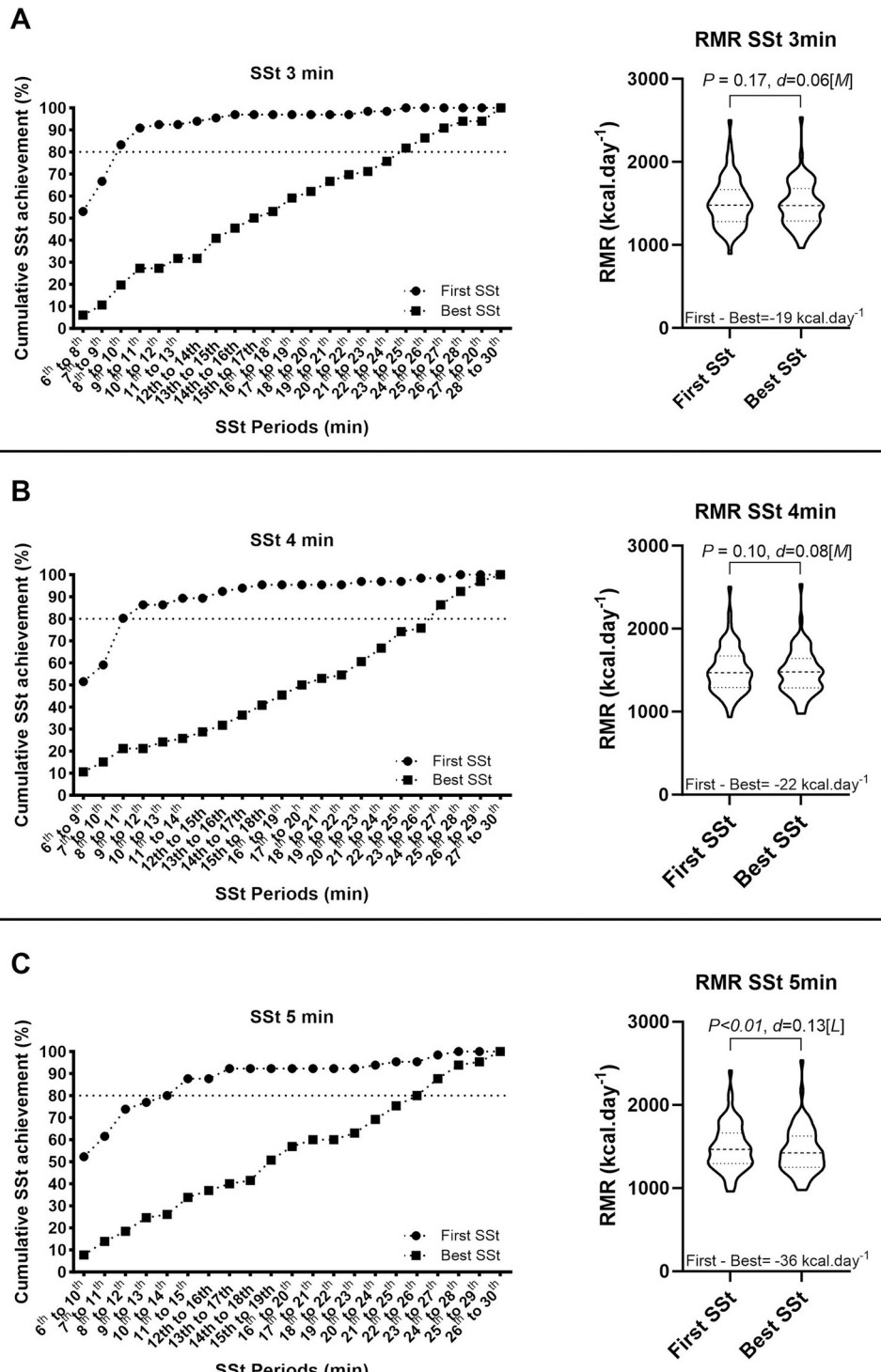

**Fig 3. The period from the resting metabolic rate assessment in which the First Steady-State (SSt) was achieved for 3 min SSt (A), 4 min SSt (B), and 5 min SSt (C).** The First SSt was achieved when the first period in which the coefficients of variation (CV) for volume of oxygen consumption (VO2) and volume of carbon dioxide production (VCO2) were lower than 10% and the CV for respiratory exchange ratio (RER) was lower than 5%. Best SSt represents the SSt period presenting the lowest mean of the CVs as mentioned. Boxes represent the cumulative percentage for the Best SSt; circles represent the cumulative percentage for the First SSt. Effect size: S—Small, M—Medium, L—Large.

work capacity in clinical trials and population studies. The estimated RMR was affected by the method of gas exchange data selection, in which the lowest value was obtained by the 5-min SSt method. The lowest CV among the gas exchange data was obtained by the SSt methods, compared to TI and Filtering, making this method more reliable for population studies. It was also tested whether methods were better associated with its classical predictor's parameters (i.e., sex, body mass and/or body composition), being observed similar correlations between RMR and the parameters among methods. Finally, as the 5-min SSt method provided the lowest RMR estimations, and no differences were obtained between the first and the best 5-min SSt estimations, the RMR assessment in young athletes could be reduced to 10 min, improving the time spent in the experimental procedures by CI.

In contrast to previous studies on young and middle-aged healthy adults [18] and high-level athletes [16], our results showed few differences among the gas exchange data selection methods (F = 2.607, $p$ = 0.004). Similarly to Alcantara et al. (2020) [18], the 5-min SSt method yielded the lowest mean RMR value (1454 ± 282 kcal.day$^{-1}$) among all the methods. Although RMR was significantly different between the methods tested, this difference could be considered clinically irrelevant for athletes' evaluations, since it was less than 46 kcal.day$^{-1}$. Considering the high metabolic rate and energy expenditure during exercise, this amount of calories during resting is inside an acceptable range of variation for the estimative of daily energy expenditure of athletes [1]. In addition, when the RMR was estimated using simpler classical predictors (i.e., body mass, Model 1), it yielded lower $R^2$ values ($R^2$ = 0.37–0.52) than compared to more complex models, including sex and body composition variables (Models 2 and 3, $R^2$ = 0.59–0.67). In the present study, we found higher $R^2$ values than those of Alcantara et al. (2020) [18] and lower than those of Freire et al. (2021) [16]. To a certain extent, it may be explained by the differences regarding the sample's age (young and middle-aged adults *vs* adolescents) and the different metabolic carts used in each study. The SSt and filtering methods provided acceptable CVs (CV<10% for VO$_2$ and VCO$_2$ and <5% for RER) [14] for most subjects. Similar results were found by Alcantara et al. (2020) [18] and by Freire et al. (2021) [16], indicating that these methods offered more "stable" values compared to long or short TI methods (i.e., less variable, as they presented lower CVs). Although these thresholds (CVs) were established based on intensive care unit patients [19], the results of the present and previous studies suggest that the limits of variance appear to be adequate for other populations also (e.g., young athletes).

To the best of our knowledge, the present study is the first to compare the influence of different gas exchange data selection methods on RMR estimation in young athletes. In a cohort of young adult athletes, Freire et al. (2021) [16] reported statistical differences between the high filter and all other methods ($p$ <0.001) without differences among the filter methods. Furthermore, Alcantara et al. (2020) [18] (who first proposed the filtering method) found similar results in two different age groups (young adults and middle-aged adults). In the middle-aged cohort, they observed that the high filter method provided the lowest RMR value and differed from almost all methods except 3, 4, and 5-min SSt and TI 11–15 min [18]. On the other hand, in the young adult cohort, they observed differences between the 4-min SSt method and the 6-10- and 11-15-minute TI methods [18]. Conversely, in our study, the filter methods did not differ from other methods, except for the low filter compared to the TI 6–30 min ($p$ = 0.028), in which the mean difference was only 6 kcal.day$^{-1}$.

The 5-min SSt has been widely considered the reference method for estimating the RMR, even considering different populations [1, 14, 19, 28, 29], but no study was found indicating the best method for young athletes. Borges et al. (2019) [28] compared different gas exchange data selection methods in a cohort of 30 adults (15 women). They concluded that the 5-min SSt method, derived from a 30-minute RMR assessment, provided the lowest RMR value.

These findings partially corroborate our results since, from the 14 methods analysed, the 5-min SSt method yielded the lowest mean RMR value. Additionally, according to their results, when RMR was analysed in the first 10-minute period, it was observed that the RMR values were overestimated by ~100 kcal.day$^{-1}$. Indeed, in our study, even after 30 minutes for acclimation, approximately only 50% obtained a valid RMR in 10 minutes of measurement, regardless of the length of data selection (i.e., SSt 3, 4, and 5min; see Fig 3).

The TI method seems to be the most practical method for determining RMR as it could be considered more straightforward than the other methods. The present study showed no statistical difference among the TI methods (e.g., fixed 5-min windows). These results make it possible to raise the discussion of reducing the total assessment time without changing the RMR values. Considering that the first period (i.e., 6–10 min TI) was not statistically (or clinically) different from the last period (i.e., 26–30 min TI), it is possible to suggest that only 10 min of measurement (after 30 min of acclimation before the test) would be reliable to obtain an accurate RMR assessment. However, the TI method showed the highest CV among the methods, suggesting some caution about its use. In fact, for some subjects, TI methods exceeded the recommended CV thresholds (see Fig 2), suggesting that some "artificial noise" (e.g., metabolic cart artefacts) could be included in the RMR estimations when using this method.

On the other hand, to identify a possible reduced protocol for RMR assessments, Alcantara et al. (2020) [18] proposed identifying, in young and middle-age adults, the First and Best SSt to test whether they influenced the RMR estimations. The First SSt is considered the first period, which accomplished the steady-state criteria (CV <10% for VO$_2$ and VCO$_2$ and <5% for RER). Instead, the Best SSt is achieved when the gas exchange parameters present the lowest variability (i.e., the lowest mean CVs). In their study, it was observed that 80% of the subjects achieved their first SSt after more than 20 min, whereas in the study by Freire et al. (2021) [16], they observed that 80% of the subjects (young adults, high-level athletes) achieved their first RMR in ~11 min (without including an acclimation period before the gas exchange measurement). In contrast to Alcantara et al. (2020) [18], the present study showed that approximately 10 min are needed to achieve the First SSt (after 30 minutes of acclimation), and no statistical differences were observed when comparing the First SSt to the Best SSt (when using the 3 and 4 min SSt methods, see Fig 3) or clinically irrelevant when considered the 5-min SSt method.

Determining the best method to estimate RMR is vital for developing proper nutritional prescriptions for young athletes [9]. Despite the high cost and necessity of trained professionals, IC is the best method for evaluating RMR. Predictive equations would be an alternative for estimating RMR, but unfortunately, most perform poorly in adolescent non-athletes [30, 31] and adolescent athletes [5, 32]. IC assessments in athletes are relatively complex because of the duration of the current protocols (~60 min if the acclimation period prior to the RMR assessment is considered) and their training and competition schedules [27, 33]. According to the present study's results, only 10 min of measurement (after an acclimation period) using the 5-min SSt method would be necessary to obtain an accurate RMR assessment. This shortening of the IC procedures and the RMR assessment may reduce the total cost, as it could speed up data collection, including more individuals in less time and with less effort. Importantly, this reduction in the measurement duration did not influence the accuracy of the RMR estimations to a greater or lesser extent, as suggested by our results (i.e., no clinically relevant differences between the first *vs* the best SSt RMR estimations).

It is also important to highlight the real need to assess RMR in adolescent athletes. It is known that the kinetics of VO2 and VCO2 can be influenced by health status, gender, and age [20]. Therefore, it is important to identify the optimal method for selecting gas exchange data to estimate RMR in this young population. A method that reduces intra-assessment variability, reflected by lower CVs for VO2, VCO2, RER, and RMR, would minimize the risk of

methodological error and enhance the accuracy of RMR estimation. Considering that RMR is the main component of total energy expenditure, it is necessary to establish differentiated techniques for gas exchange analysis for this measurement.

This study has some limitations. Although the subjects' age suggests they were at the end of the maturation phase, it was not possible to assess this condition for logistical and ethical reasons. Since maturation plays an important role in adolescents' physiological systems, RMR may be affected to a certain extent [34]. However, this study did not compare participants. Thus, the possible influence of the maturation status could be reduced. In addition, the last meal was not controlled. Despite the 8–12 h of fasting by all subjects (which is recommended by the current guidelines [14], the previous meal composition may affect RMR [14]. However, it should be noted that previous literature has suggested that more than one day is necessary to influence fasting substrate utilisation [35–37]. Data acquisition method (one sample per minute) may increase methodological error and it is necessary to evaluate the methods of selecting gas exchange analysis further using breath-by-breath sample collection method. Finally, it was not possible to determine the inter-day reproducibility of the RMR. Thus, whether the method influences inter-day reproducibility in this cohort of adolescent athletes remains elusive.

In conclusion, the 5-min SSt method should be employed for assessing the RMR among young athletes, considering the possibility of obtaining a shortened assessment (~12 min) with an accepted and low coefficient of variation.

## Supporting information

**S1 Checklist. STROBE statement—Checklist of items that should be included in reports of observational studies.**
(DOCX)

**S1 Data.**
(XLSX)

## Acknowledgments

The authors thank all the athletes who voluntarily participated in this study.

## Author Contributions

**Conceptualization:** Victor Zaban Bittencourt, Raul Freire, Luiz Lannes Loureiro, Fábio Luiz Candido Cahuê, Anna Paola Trindade Rocha Pierucci.

**Data curation:** Taillan Martins de Oliveira.

**Formal analysis:** Luiz Lannes Loureiro.

**Funding acquisition:** Anna Paola Trindade Rocha Pierucci.

**Investigation:** Taillan Martins de Oliveira.

**Methodology:** Raul Freire, Juan M. A. Alcantara.

**Supervision:** Alex Itaborahy, Anna Paola Trindade Rocha Pierucci.

**Writing – original draft:** Victor Zaban Bittencourt, Raul Freire, Juan M. A. Alcantara, Fábio Luiz Candido Cahuê, Anna Paola Trindade Rocha Pierucci.

**Writing – review & editing:** Juan M. A. Alcantara, Luiz Lannes Loureiro, Fábio Luiz Candido Cahuê, Alex Itaborahy, Anna Paola Trindade Rocha Pierucci.

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
