## [Decision Letter · Decision Letter 0]

14 Jun 2023

PONE-D-23-08324Effect of gas exchange data selection methods on resting metabolic rate estimation in young athletesPLOS ONE

Dear Dr. Pierucci,

Thank you for submitting your manuscript to PLOS ONE. After careful consideration, we feel that it has merit but does not fully meet PLOS ONE’s publication criteria as it currently stands. Therefore, we invite you to submit a revised version of the manuscript that addresses the points raised during the review process.

We look forward to receiving your revised manuscript.

Kind regards,

Renato S. Melo, PhD

Academic Editor

PLOS ONE

Journal Requirements:

“This research received funding from FAPERJ. J.M.A. is supported by Grant FJC2020-044453-I funded by MCIN/AEI/ 10.13039/501100011033 and by "European Union NextGenerationEU/PRTR" and A.P.T.R.P. is supported by Grant "Cientista do Nosso Estado", from FAPERJ.”

Additional Editor Comments (if provided):

Dear Dr. Anna Paola T.R. Pierucci

The reviewers finalized their manuscript assessment, and suggested major revisions. Please complete all requested changes as quickly as possible, or create a letter to reviewers explaining your justifications for rebuttals. Highlight what was changed in the manuscript in the text in yellow, to make it easier for reviewers to more quickly identify what was changed in the manuscript.

Cordial hugs,

Reviewers' comments:

Reviewer's Responses to Questions

**Comments to the Author**

1. Is the manuscript technically sound, and do the data support the conclusions?

Reviewer #1: Yes

Reviewer #2: Yes

2. Has the statistical analysis been performed appropriately and rigorously? 

Reviewer #1: Yes

Reviewer #2: I Don't Know

3. Have the authors made all data underlying the findings in their manuscript fully available?

Reviewer #1: Yes

Reviewer #2: Yes

4. Is the manuscript presented in an intelligible fashion and written in standard English?

Reviewer #1: Yes

Reviewer #2: Yes

5. Review Comments to the Author

Reviewer #1: The authors assessed several processing techniques for a resting metabolic rating assessment in youth athletes. The study presents good quality, but some adjustments are necessary to improve readers' understanding and promotion of the best practices of RMR assessment for this population. Manuscript improvements are suggested according to session and line counting from the submission file. I would like to congratulate the authors and encourage them to invest in these enhancements to enrich the quality of their study presentation.

-Abstract-

Q1. In the objectives, the authors mention "shortening protocol". I believe the authors meant protocol shortening. Please, check it.

-Introduction-

Q2. [Page 3 - Lines 50-52]: This sentence sounds redundant. "…have increased energy needs than sedentary individuals because they have greater energy demands." Please, rephrase or be more specific.

Q3. [Page 3 – Line 60]: A period (full stop) is missing after Weir, 1949. Please, check it.

Q4. [Introduction – 2nd paragraph]: The second paragraph is too long. I suggest breaking it down into 2 or 3 paragraphs. Maybe, lines 65 and 81 may be split into new paragraphs. Please, reconsider the length.

Q5. [Overall Introduction]: The Introduction is well written and gathers good evidence to lead authors to the aimed gap. However, it is still unclear why the sample effect would affect the signal nature and confirm the originality besides the previous studies that have already compared the different techniques. Please, consider improving the gap elucidation.

-Methods-

Q6. [Page 5 - Lines 96-97]: 12 to 18 years is a wide gap. Is any assessment of maturation included in sample profiling?

Q7. [Page 6 – Line 132-133]: Was the equipment set to collect every minute, or was a fixed time averaging of 1 min was set as standard output? Please, specify it.

Q8. [Page 6 – Line 148-149]: "(…) values within the filtering intensities", you mean that it met the thresholds or ranges, right? Please, adjust it.

Q9. [Page 7 – Line 150-151]: Instead of "it was computed the first…", please adjust the sentence like "The first and the best SSt were computed as previously proposed…".

Q10. [Page 7 – Line 150-156]: Please, gather all SSt detail in the same paragraph (e.g., between TI and Filtering or after Filtering).

Q11. [Page 7 – Line 155-156]: Please, cite the Weir equation reference in addition to Hansen et al. (1998).

Q12. [Page 8 – Line 162-165]: Could authors further explain this ANOVA choice and the invalid values issues? It was not clear, and readers may not follow these missing values.

-Results-

Q13. [Table 1]: Please, replace all decimal commas with dots and standardize the decimal places for each variable.

Q14. [Figures 1 and 2]: Please include the effect sizes for all comparisons.

Q15. [Figures 1 and 2 – and Methods]: It was unclear if the authors included the Best or First SST methods in this analysis. I needed to read several times the methods section, and it is still not totally clear after it was broken down for Figure 3. Were methods combined here? Please, improve the explanation or include another name for the combined approach.

Q16. [Page 9 – Line 187]: I believe the authors meant to cite subfigure C instead of B. Please, adjust it.

Q17. [Page 9 – Lines 193-194]: I believe the authors meant to cite subfigure B instead of C. Please, adjust it.

Q18. [Page 9 – Lines 205-208]: Figure 2 only presents CVs' mean (sd). The occurrence of the achievement of CV limits was not plotted. Please, consider including or replacing it with a dispersion illustration highlighting this occurrence or reconsider the sentence.

Q19. [Page 10 – Line 215]: There is an incomplete parathesis between "SEE" and "of approximately…". Please, adjust it.

Q20. [Page 10 – Line 217]: You have cited Table 3 twice. Please, adjust it.

Q21. [Page 10 – Line 217]: I think the authors missed the "Figure 3 Here" indication after the last results paragraph.

-Discussion-

Q22. [Overall Discussion]: Please, review the citing style [e.g., In contrast to Alcantara et al. (Alcantara et al., 2020), (...)]. It seems redundant and repeats several times across the paragraphs. In Line 272, brackets are used "(…) Freire et al. [14]".

Q23. [Discussion - 2nd and 3rd Paragraphs]: Please, elucidate how populations may interfere with the signal nature. In addition, I am concerned if differences can be attributed to populations solely or the metabolic carts brand and, maybe more determinately, the signal sampling adopted.

Q24. [Discussion – 6th paragraph]: Please, also mention the populations when comparing the studies to improve readers' understanding.

Q25. [Discussion – 8th Paragraph]: The signal sampling of 1 min can also limit the assessment of the methods. Please, consider discussing it.

Reviewer #2: Dear Authors,

I would like to congratulate you for your work. The method and procedure of the study make the study worthwhile to be published in this journal. I have some recommendations for you.

First of all, in the beginning of method section you mentioned 5 different sports but there is only for, please rewrite it.

Second, I think you need an English proofreading support.

Best Regards

6. PLOS authors have the option to publish the peer review history of their article (what does this mean?). If published, this will include your full peer review and any attached files.

Reviewer #1: No

Reviewer #2: No

---

## [Author Response · Author response to Decision Letter 0]

26 Jul 2023

Dear editors and reviewers., 

The authors would like to thank you for these much relevant comments and suggestions, which contribute to improving the quality of our paper. After meticulously analyzing the comments and questionings, the errors that were pointed out, and the suggestions that were sent to us, we modified it to improve the writing and the overall understanding of our paper. We hereby send the explanations and modifications related to the comments we received. All modifications in the manuscript were tagged in yellow in file with track changes.

Abstract

Q1. In the objectives, the authors mention "shortening protocol". I believe the authors meant protocol shortening. Please, check it.

A: Thank you. We modify it following your consideration.

Introduction

Q2. [Page 3 - Lines 50-52]: This sentence sounds redundant. "…have increased energy needs than sedentary individuals because they have greater energy demands." Please, rephrase or be more specific.

A: We modified it to be more specific to energy needs due to exercise training. The sentence, after modification, is “...have higher energy needs than sedentary individuals because training promotes higher energy demands”.

Q3. [Page 3 – Line 60]: A period (full stop) is missing after Weir, 1949. Please, check it.

A: We insert a period after Weir, 1949. Thank you.

Q4. [Introduction – 2nd paragraph]: The second paragraph is too long. I suggest breaking it down into 2 or 3 paragraphs. Maybe, lines 65 and 81 may be split into new paragraphs. Please, reconsider the length.

A: As suggested, we splitted the second paragraph in 3.

Q5. [Overall Introduction]: The Introduction is well written and gathers good evidence to lead authors to the aimed gap. However, it is still unclear why the sample effect would affect the signal nature and confirm the originality besides the previous studies that have already compared the different techniques. Please, consider improving the gap elucidation.

A: Thank you for your question. We insert the following sentence in the introduction. “...given that it is known that VO2 and VCO2 kinetics could be influenced by gender, age and health status (Alcantara et al., 2022)”

Methods

Q6. [Page 5 - Lines 96-97]: 12 to 18 years is a wide gap. Is any assessment of maturation included in sample profiling?

A: As the visual assessment using Tanner's Maturation Scale is considered the most reliable protocol for evaluating maturation, and due to ethical considerations in Brazil requiring assessment by a Pediatric Doctor, we did not include maturation evaluation in our samples. We considered this as a limitation in the last paragraph of the discussion.

Q7. [Page 6 – Line 132-133]: Was the equipment set to collect every minute, or was a fixed time averaging of 1 min was set as standard output? Please, specify it.

A: The equipment collected one sample each 1 minute. We describe it in Methods.

Q8. [Page 6 – Line 148-149]: "(…) values within the filtering intensities", you mean that it met the thresholds or ranges, right? Please, adjust it.

A: We adjusted the text to make it clearer.

Q9. [Page 7 – Line 150-151]: Instead of "it was computed the first…", please adjust the sentence like "The first and the best SSt were computed as previously proposed…".

A: It was adjusted as suggested.

Q10. [Page 7 – Line 150-156]: Please, gather all SSt detail in the same paragraph (e.g., between TI and Filtering or after Filtering).

A: It was adjusted as suggested.

Q11. [Page 7 – Line 155-156]: Please, cite the Weir equation reference in addition to Hansen et al. (1998).

A: We exchanged Hansen et al. by Weir et al..

Q12. [Page 8 – Line 162-165]: Could authors further explain this ANOVA choice and the invalid values issues? It was not clear, and readers may not follow these missing values.

A: Possibly there was a confounding use of statistical terms here. GraphPad Prism calls Mixed ANOVA when there are missing values in the matrix for ANOVA’s calculations. In this case, the software fitting the data to insert the “missing” values and run the repeated measures ANOVA. This strategy was used instead of excluded the valid data (calculated using other data selection methods) for the subject. Only TI6-10 and 10minSSt returned missing data and we reported it in the text to let it clear to the reader. We preferred to exclude the term “mixed” in the text to eliminate a possible confusion about the statistical terms. Finally, please see the following link link (https://www.graphpad.com/guides/prism/latest/statistics/stat_missing-values-in-repeated-mea.htm) with the explanation regarding the missing values in repeated measures ANOVA.

Results

Q13. [Table 1]: Please, replace all decimal commas with dots and standardize the decimal places for each variable.

A: The values were adjusted as suggested. 

Q14. [Figures 1 and 2]: Please include the effect sizes for all comparisons.

A: The effect sizes were included as suggested.

Q15. [Figures 1 and 2 – and Methods]: It was unclear if the authors included the Best or First SST methods in this analysis. I needed to read several times the methods section, and it is still not totally clear after it was broken down for Figure 3. Were methods combined here? Please, improve the explanation or include another name for the combined approach.

A: We made a minor adjustment in the statistical section trying to let it clearer.

Q16. [Page 9 – Line 187]: I believe the authors meant to cite subfigure C instead of B. Please, adjust it.

A: Thank you. It was amended.

Q17. [Page 9 – Lines 193-194]: I believe the authors meant to cite subfigure B instead of C. Please, adjust it.

A: Thank you. It was amended.

Q18. [Page 9 – Lines 205-208]: Figure 2 only presents CVs' mean (sd). The occurrence of the achievement of CV limits was not plotted. Please, consider including or replacing it with a dispersion illustration highlighting this occurrence or reconsider the sentence.

A: We made a minor adjustment in the statistical section trying to let it clearer.

Q19. [Page 10 – Line 215]: There is an incomplete parathesis between "SEE" and "of approximately…". Please, adjust it.

A: Thank you. It was adjusted.

Q20. [Page 10 – Line 217]: You have cited Table 3 twice. Please, adjust it.

A: Thank you. It was adjusted.

Q21. [Page 10 – Line 217]: I think the authors missed the "Figure 3 Here" indication after the last results paragraph.

A: Thank you. It was adjusted.

Discussion

Q22. [Overall Discussion]: Please, review the citing style [e.g., In contrast to Alcantara et al. (Alcantara et al., 2020), (...)]. It seems redundant and repeats several times across the paragraphs. In Line 272, brackets are used "(…) Freire et al. [14]".

A: Thank you. It was adjusted in the whole discussion.

Q23. [Discussion - 2nd and 3rd Paragraphs]: Please, elucidate how populations may interfere with the signal nature. In addition, I am concerned if differences can be attributed to populations solely or the metabolic carts brand and, maybe more determinately, the signal sampling adopted.

A: Thank you for your question. We wrote a new paragraph before 8th paragraph elucidating how populations may interfere signal nature and, thus, RMR estimation. 

“It is also important to highlight the real need to assess RMR in adolescent athletes. It is known that the kinetics of VO2 and VCO2 can be influenced by health status, gender, and age (Alcantara et al., 2022). Therefore, it is important to identify the optimal method for selecting gas exchange data to estimate RMR in this young population. A method that reduces intra-assessment variability, reflected by lower CVs for VO2, VCO2, RER, and RMR, would minimize the risk of methodological error and enhance the accuracy of RMR estimation. Considering that RMR is the main component of total energy expenditure, it is necessary to establish differentiated techniques for gas exchange analysis for this measurement.“

Q24. [Discussion – 6th paragraph]: Please, also mention the populations when comparing the studies to improve readers' understanding.

A: Thank you. It was adjusted.

Q25. [Discussion – 8th Paragraph]: The signal sampling of 1 min can also limit the assessment of the methods. Please, consider discussing it.

A: Thank you. The sentence below was added in 9th (old 8th) paragraph.

“Data acquisition method (one sample per minute) may increase methodological error and it is necessary to evaluate the methods of selecting gas exchange analysis further using breath-by-breath sample collection method.

---

## [Decision Letter · Decision Letter 1]

21 Aug 2023

PONE-D-23-08324R1Effect of gas exchange data selection methods on resting metabolic rate estimation in young athletesPLOS ONE

Dear Dr. Pierucci,

Thank you for submitting your manuscript to PLOS ONE. After careful consideration, we feel that it has merit but does not fully meet PLOS ONE’s publication criteria as it currently stands. Therefore, we invite you to submit a revised version of the manuscript that addresses the points raised during the review process.

We look forward to receiving your revised manuscript.

Kind regards,

Renato S. Melo, PhD

Academic Editor

PLOS ONE

Journal Requirements:

Additional Editor Comments:

Dear Author, We have received the reviewers' careful evaluations of your manuscript from the reviewers, however, however, the reviewers have requested some changes to your manuscript.

Reviewers' comments:

Reviewer's Responses to Questions

**Comments to the Author**

1. If the authors have adequately addressed your comments raised in a previous round of review and you feel that this manuscript is now acceptable for publication, you may indicate that here to bypass the “Comments to the Author” section, enter your conflict of interest statement in the “Confidential to Editor” section, and submit your "Accept" recommendation.

Reviewer #1: All comments have been addressed

Reviewer #2: All comments have been addressed

2. Is the manuscript technically sound, and do the data support the conclusions?

Reviewer #1: Yes

Reviewer #2: Yes

3. Has the statistical analysis been performed appropriately and rigorously? 

Reviewer #1: Yes

Reviewer #2: I Don't Know

4. Have the authors made all data underlying the findings in their manuscript fully available?

Reviewer #1: Yes

Reviewer #2: Yes

5. Is the manuscript presented in an intelligible fashion and written in standard English?

Reviewer #1: Yes

Reviewer #2: Yes

6. Review Comments to the Author

Reviewer #1: I would like to congratulate the authors for their effort in addressing all the queries that were pointed out. As previously mentioned, the paper has great merit but requires some enhancements before publication. While improvements have been made to the paper, a few adjustments are still needed to meet the desired standards.

*Q8. [Page 6 – Line 148-149]: "(…) values within the filtering intensities", you mean that

it met the thresholds or ranges, right? Please, adjust it.

A: We adjusted the text to make it clearer.

**Q8.R2 – The adjustment was suitable. Please, correct the Alcantara parenthesis citation.

*Q13. [Table 1]: Please, replace all decimal commas with dots and standardize the decimal places for each variable.

A: The values were adjusted as suggested.

**Q13.R2. The numbers still have commas. I believe there was a missed saving or file mistake with the reviewed version. Please replace them.

*Q14. [Figures 1 and 2]: Please include the effect sizes for all comparisons.

A: The effect sizes were included as suggested.

**Q14.R2. Thank you for including ES. However, eta-squared thresholds classification is still missing.

*Q18. [Page 9 – Lines 205-208]: Figure 2 only presents CVs' mean (sd). The

occurrence of the achievement of CV limits was not plotted. Please, consider including

or replacing it with a dispersion illustration highlighting this occurrence or reconsider

the sentence.

A: We made a minor adjustment in the statistical section trying to let it clearer.

**Q18.R2. Thank you for adjusting it. A subject is still missing in this sentence. I guess the authors removed the “75% of the subjects (3rd quartile)” since the plot only presents meanDP. However, the corrected sentence remained incomplete. Please adjust it.

I believe that after those few minor corrections, the paper will meet the standards of PlosOne. I would like to thank the authors for their diligence during the review process.

Reviewer #2: Dear Authors,

I would like to thank you for your effort. I am sure the paper will contribute to the literature.

Regards

7. PLOS authors have the option to publish the peer review history of their article (what does this mean?). If published, this will include your full peer review and any attached files.

Reviewer #1: No

Reviewer #2: No

---

## [Author Response · Author response to Decision Letter 1]

29 Aug 2023

Answers

*Q8. [Page 6 – Line 148-149]: "(…) values within the filtering intensities", you mean that

it met the thresholds or ranges, right? Please, adjust it.

A: We adjusted the text to make it clearer.

**Q8.R2 – The adjustment was suitable. Please, correct the Alcantara parenthesis citation.

A: It was adjusted.

*Q13. [Table 1]: Please, replace all decimal commas with dots and standardize the decimal places for each variable.

A: The values were adjusted as suggested.

**Q13.R2. The numbers still have commas. I believe there was a missed saving or file mistake with the reviewed version. Please replace them.

A: It was adjusted.

*Q14. [Figures 1 and 2]: Please include the effect sizes for all comparisons.

A: The effect sizes were included as suggested.

**Q14.R2. Thank you for including ES. However, eta-squared thresholds classification is still missing.

A: The effect size thresholds classification were included in figures and the legend (S-Small, M-Medium and L-Large) were described in Statistical Analysis and in each figure’s caption.

*Q18. [Page 9 – Lines 205-208]: Figure 2 only presents CVs' mean (sd). The

occurrence of the achievement of CV limits was not plotted. Please, consider including

or replacing it with a dispersion illustration highlighting this occurrence or reconsider

the sentence.

A: We made a minor adjustment in the statistical section trying to let it clearer.

**Q18.R2. Thank you for adjusting it. A subject is still missing in this sentence. I guess the authors removed the “75% of the subjects (3rd quartile)” since the plot only presents meanDP. However, the corrected sentence remained incomplete. Please adjust it.

A: The sentence was adjusted.

---

## [Decision Letter · Decision Letter 2]

31 Aug 2023

Effect of gas exchange data selection methods on resting metabolic rate estimation in young athletes

PONE-D-23-08324R2

Dear Dr. Pierucci,

We’re pleased to inform you that your manuscript has been judged scientifically suitable for publication and will be formally accepted for publication once it meets all outstanding technical requirements.

Kind regards,

Renato S. Melo, PhD

Academic Editor

PLOS ONE

Reviewers' comments:

Reviewer's Responses to Questions

**Comments to the Author**

1. If the authors have adequately addressed your comments raised in a previous round of review and you feel that this manuscript is now acceptable for publication, you may indicate that here to bypass the “Comments to the Author” section, enter your conflict of interest statement in the “Confidential to Editor” section, and submit your "Accept" recommendation.

Reviewer #1: All comments have been addressed

2. Is the manuscript technically sound, and do the data support the conclusions?

Reviewer #1: Yes

3. Has the statistical analysis been performed appropriately and rigorously? 

Reviewer #1: Yes

4. Have the authors made all data underlying the findings in their manuscript fully available?

Reviewer #1: Yes

5. Is the manuscript presented in an intelligible fashion and written in standard English?

Reviewer #1: Yes

6. Review Comments to the Author

Reviewer #1: All requested corrections were made. Congratulations to the authors for their work. The manuscript is ready for publication according to PlosOne standards.

7. PLOS authors have the option to publish the peer review history of their article (what does this mean?). If published, this will include your full peer review and any attached files.

Reviewer #1: No

---

## [Editor Report · Acceptance letter]

11 Sep 2023

PONE-D-23-08324R2 

Effect of gas exchange data selection methods on resting metabolic rate estimation in young athletes 

Dear Dr. Pierucci:

I'm pleased to inform you that your manuscript has been deemed suitable for publication in PLOS ONE. Congratulations! Your manuscript is now with our production department. 

Kind regards, 

on behalf of

Dr. Renato S. Melo 

Academic Editor

PLOS ONE